# You Only Submit One Image to Find the Most Suitable Generative Model

## Abstract

Deep generative models have achieved promising results in image generation, and various generative model hubs, e.g., Hugging Face and Civitai, have been developed that enable model developers to upload models and users to download models. However, these model hubs lack advanced model management and identification mechanisms, resulting in users only searching for models through text matching, download sorting, etc., making it difficult to efficiently find the model that best meets user requirements. In this paper, we propose a novel setting called *Generative Model Identification* (GMI), which aims to enable the user to identify the most appropriate generative model(s) for the user's requirements from a large number of candidate models efficiently. To our best knowledge, it has not been studied yet. In this paper, we introduce a comprehensive solution consisting of three pivotal modules: a weighted Reduced Kernel Mean Embedding (RKME) framework for capturing the generated image distribution and the relationship between images and prompts, a pre-trained vision-language model aimed at addressing dimensionality challenges, and an image interrogator designed to tackle cross-modality issues. Extensive empirical results demonstrate the proposal is both efficient and effective. For example, users only need to submit a single example image to describe their requirements, and the model platform can achieve an average top-4 identification accuracy of more than 80%. The code and benchmark are all released to promote the research.

## 1 Introduction

Deep generative models (Jebara, 2012), including variational autoencoder (VAE) (Kingma & Welling, 2014; 2019; Parmar et al., 2021), generative adversarial network (GAN) (Creswell et al., 2018; Mirza & Osindero, 2014; Sohn et al., 2015), flow-based model (Kobyzev et al., 2021; Rezende & Mohamed, 2015), and the diffusion models (Dhariwal & Nichol, 2021; Sohl-Dickstein et al., 2015), have achieved remarkable performance in image generation. Recently, stable diffusion models (Rombach et al., 2022) have achieved state-of-the-art generative capabilities and become one of the popular topics in artificial intelligence. Various model hubs, e.g., Hugging Face[1] and Civitai[2], have been developed to enable model developers to upload and share their generative models.

Existing model hubs provide some trivial methods such as tag filtering, text matching, and download volume ranking (Shen et al., 2023), to help users search for models. However, these methods cannot accurately capture the users' requirements, making it difficult to efficiently identify the most appropriate model for users. As shown in Figure 1, the user should submit their requirements (usually in text) to the model hub and subsequently, they must download and evaluate the searched model one by one until they find the satisfactory one, causing significant time and computing resources.

The above limitation of existing generative model hubs inspires us to consider the following question: Can we describe the functionalities and utilities of different generative models more precisely in some format that enables the model can be efficiently and accurately identified in the future by matching the models' functionalities with users' requirements? We call this novel setting *Generative Model Identification* (GMI). To the best of our knowledge, this problem has not been studied yet.

---

[1]https://huggingface.co/
[2]https://civitai.com/

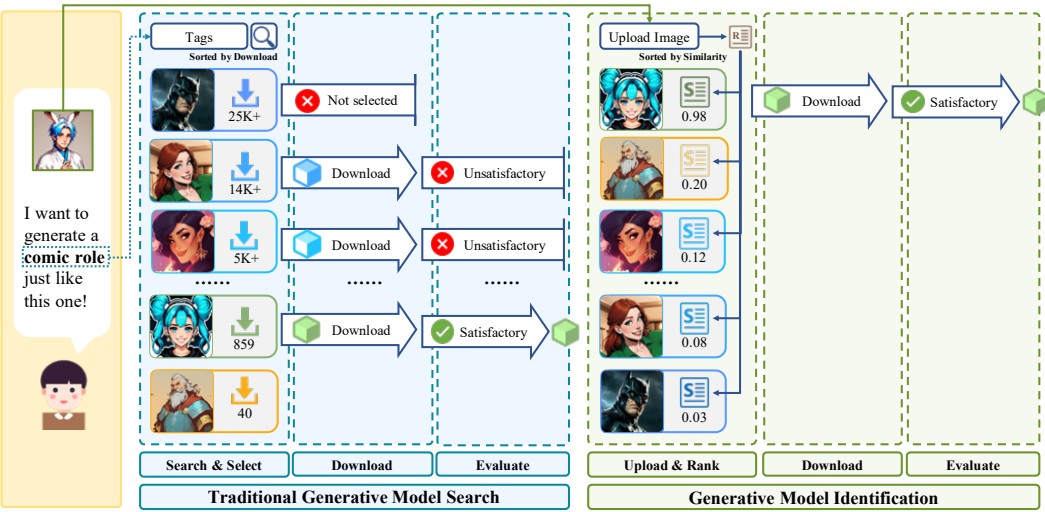

Figure 1: Comparison between traditional generative model search of existing model hubs and GMI. GMI matches requirements and specifications during the identification process. A detailed explanation is presented in B.

It is evident that two problems need to be addressed to achieve GMI, the first is how to describe the functionalities of different generative models, and the second is how to match the user requirements with the models' functionalities. Inspired by the learnware paradigm (Zhou, 2016), which proposes to assign a specification to each model that reflects the model's utilities, we adopt the Reduced Kernel Mean Embedding (RKME) as the model specification to capture the distribution of generated images produced by different generative models, since the generated image distribution could reflect the model functionality. However, previous RKME studies mainly focus on classification tasks, and can not be directly applied to generative models. To this end, we propose a novel systematic solution consisting of three pivotal modules: a weighted Reduced Kernel Mean Embedding (RKME) framework for capturing not only the generated image distribution but also the relationship between images and prompts, a pre-trained vision-language model aimed at addressing dimensionality challenges, and an image interrogator designed to tackle cross-modality issues. For the second problem, we assume the user can present one image as an example to describe the requirements, and then we can match the model specification with the example image to compute how well each candidate generative model matches users' requirements. Figure 1 provides a comparison between previous model search methods and the new solution. The goal is to identify the most suitable generative model with only one single image as an example to describe the user's requirements.

To evaluate the effectiveness of our proposal, we construct a benchmark platform consisting of 16 tasks specifically designed for GMI using stable diffusion models. The experiment results show that our proposal is both efficient and effective. For example, users only need to submit a single example image to describe their requirements, and the model platform can achieve an average top-4 identification accuracy of more than 80%, indicating that recommending four models can satisfy users in major cases on the benchmark dataset.

## 2 PROBLEM AND ANALYSIS

In this section, we first describe the notation and formulation of GMI. Then, we theoretically discuss the obstacles existing methods face in generative models. Finally, we propose an advanced formulation motivated by our analysis.

### 2.1 PROBLEM SETUP

In this paper, we explore a novel problem setting called GMI, where users identify the most appropriate generative models for their specific purposes using just one single image. We assume there

is a model platform, consisting of $M$ generative models $\{f_m\}_{m=1}^M$. Each model is associated with a corresponding specification $S_m$ to describe its functionalities for future model identification. The platform consists of two stages: the submitting stage for model developers and the identification stage for users.

In the submitting stage, the model developer submits a generative model $f_m$ to the platform. Then, the platform assigns a specification $S_m$ to this model. Here, the specification $S_m = \mathcal{A}_s(f_m, \mathbf{P})$ is generated by a specification algorithm $\mathcal{A}_s$ using the model $f_m$ and a prompt set $\mathbf{P} = \{\mathbf{p}_k\}_{k=1}^N$. If the model developer can provide a specific prompt set for the uploaded model, the generated specification would be more precise in describing its functionalities. In the identification stage, the users identify models from the platform using only one image $\mathbf{x}_\tau$. When users upload an image $\mathbf{x}_\tau$ to describe their purposes, the platform automatically calculates the pseudo-prompt $\widehat{\mathbf{p}}_\tau$ and then generates requirements $R_\tau = \mathcal{A}_r(\mathbf{x}_\tau, \widehat{\mathbf{p}}_\tau)$ using a requirement algorithm $\mathcal{A}_r$. Users can optionally provide corresponding prompt $\mathbf{p}_\tau$, setting $\widehat{\mathbf{p}}_r = \mathbf{p}_\tau$, to more precisely describe their purposes. During the identification process, the platform matches requirement $R_\tau$ with model specifications $\{S_m\}_{m=1}^M$ using a evaluation algorithm $\mathcal{A}_e$ and compute similarity score $\widehat{s}_{\tau,m} = \mathcal{A}_e(S_m, R_\tau)$ for each model $f_m$. Finally, the platform returns the best-matched model with the maximum similarity score or a list of models sorted by $\{\widehat{s}_{\tau,m}\}_{m=1}^M$ in descending order.

Note that the GMI setting helps reduce the consumption of network traffic and computing resources, as well as the time and effort of users. As shown in Figure 1, users are relieved from the burden of repeatedly selecting, downloading, and evaluating models by utilizing the calculated similarity scores $\{\widehat{s}_{\tau,m}\}_{m=1}^M$. Moreover, the GMI setting is easy to use for both developers and users since all the processes are automatically conducted in the background without requiring complex inputs.

There are two main challenges for addressing GMI setting: 1) In the submitting stage, how to design $\mathcal{A}_s$ to fully characterize the generative models for identification? 2) In the identification stage, how to design $\mathcal{A}_r$ and $\mathcal{A}_e$ to effectively identify the most appropriate generative models for user needs?

## 2.2 PROBLEM ANALYSIS

In this subsection, we first briefly introduce the principle of the generative model, taking the stable diffusion models as examples. Then, we show the RKME method (Wu et al., 2023) can address GMI as a baseline method, modeling the data distribution of the model as the specification. We present an example to show impossible cases of the baseline method because of overlooking the interplay between prompts and images for generative tasks. Finally, we introduce our weighted RKME framework for solving GMI problem setting.

**Stable Diffusion.** Generative models (Jebara, 2012) are capable of sampling images from a data distribution defined by the model. Recently, stable diffusion models (Dhariwal & Nichol, 2021) have become one of the most popular models for their impressive performance. Therefore, we take conditional stable diffusion models as examples for subsequent analysis and experiments. The conditional diffusion model is a latent variable model, modeling a Markov chain with learned Gaussian transitions $p_{\theta_m}(\mathbf{x}_{t-1}|\mathbf{x}_t; \mathbf{p})$ for each iteration $t \in [1, T]$ starting an initial state $p(\mathbf{x}_T) \sim \mathcal{N}(\mathbf{0}, \mathbf{I})$:

$$p_{\theta_m}(\mathbf{x}_{0:T}|\mathbf{p}) = p(\mathbf{x}_T) \prod_{t=1}^T p_{\theta_m}(\mathbf{x}_{t-1}|\mathbf{x}_t; \mathbf{p}) \tag{1}$$

Here, $\mathbf{p}$ is a prompt guiding the generation process, and $p_{\theta_m}(\cdot)$ is the learned Gaussian transitions parameterized by $\theta_m$. For simplicity, we assume the generative model $f_m$ generate an image set $\mathbf{X}_m = \{\mathbf{x}_{m,i}\}_{i=1}^N = \{\mathbf{x}|\mathbf{x} \sim f_m(\mathbf{p}), \forall \mathbf{p} \in \mathbf{P}\}$ sampled from corresponding probability distribution $p_{\theta_m}(\mathbf{x}_{0:T}|\mathbf{p})$, using prompt set $\mathbf{P}$ of model platform.

**Reduced Kernel Mean Embedding.** A baseline method to describe the model's functionality is the RKME techniques (Wu et al., 2023). It maps data distribution of each model $f_m$ as corresponding specification $S_m^{\text{RKME}} = \{\mathbf{x}_{m,i}^{\text{RKME}}\}_{i=1}^{N_m^{\text{RKME}}}$, where $N_m^{\text{RKME}}$ is the reduced set size of $f_m$. For one query image $\mathbf{x}_\tau$ from the users, the baseline method defines the requirement as $R_\tau^{\text{RKME}} = \{\mathbf{x}_\tau\}$. Finally,

the platform computes the similarity score in RKHS $\mathcal{H}_k$ using evaluation algorithm $\mathcal{A}_e^{\text{RKME}}$:

$$\mathcal{A}_e^{\text{RKME}}(S_m^{\text{RKME}}, R_\tau^{\text{RKME}}) = \left\| \sum_{i=1}^{N_m^{\text{RKME}}} \frac{1}{N_m^{\text{RKME}}} k(\mathbf{x}_{m,i}^{\text{RKME}}, \cdot) - k(\mathbf{x}_\tau \cdot) \right\|_{\mathcal{H}_k}^2 \tag{2}$$

where $k(\cdot, \cdot)$ is the reproducing kernels associated with RKHS $\mathcal{H}_k$. This baseline method fails to capture the interplay between generated images $\mathbf{X}_m$ and the prompt set $\mathbf{P}$, which is the probability distribution $p_{\theta_m}(\mathbf{x}_{0:T}|\mathbf{p})$ inside the generative model $f_m$. We present an example to show this interplay is important otherwise the specification cannot distinguish two models in specific cases, resulting in unsatisfactory identification results.

**Example 2.1.** Suppose that there are two simplified generative models $f_1$ and $f_2$ on the platform. $f_1$ generates scatter points following $x = \cos(p\pi), y = \sin(p\pi)$. $f_2$ generates scatter points following $x = \sin(p\pi), y = \cos(p\pi)$. The prompt set $\mathbf{p}$ follows $\mathcal{U}(-1,1)$. The user wants to deploy the identified model conditioned on prompts $\mathbf{p}_\tau$ following distribution $\mathcal{U}(0.5, 0)$. In Figure 2, we show that the baseline method in Equation 2 fails to distinguish two models $f_1$ and $f_2$ for the user. However, the two models function differently with $\mathbf{p}_\tau$. Figure 2a and Figure 2b show that

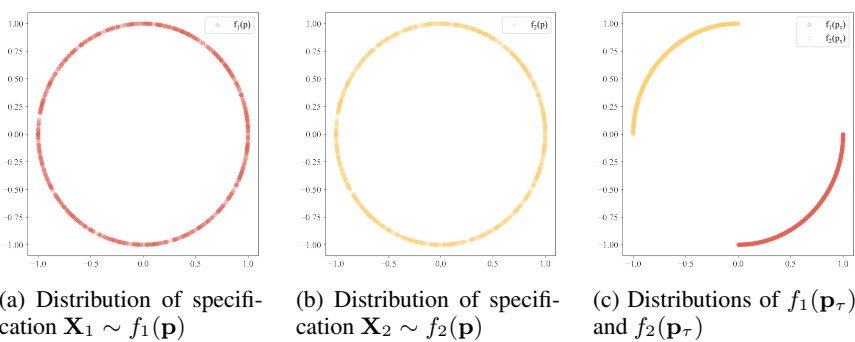

(a) Distribution of specification $\mathbf{X}_1 \sim f_1(\mathbf{p})$

(b) Distribution of specification $\mathbf{X}_2 \sim f_2(\mathbf{p})$

(c) Distributions of $f_1(\mathbf{p}_\tau)$ and $f_2(\mathbf{p}_\tau)$

Figure 2: Baseline method in Equation 2 fails to distinguish two different models for users.

although models $f_1$ and $f_2$ function differently, the data distribution $\mathbf{X}_1 \sim f_1(\mathbf{p})$ and $\mathbf{X}_2 \sim f_2(\mathbf{p})$, conditioned on the default prompt distribution $\mathbf{p}$, could be identical. Therefore, the specificaions $S_1^{\text{RKME}}$ and $S_2^{\text{RKME}}$ are identical, resulting in the same similarity scores $\mathcal{A}_e^{\text{RKME}}(S_1^{\text{RKME}}, R_\tau^{\text{RKME}})$ and $\mathcal{A}_e^{\text{RKME}}(S_2^{\text{RKME}}, R_\tau^{\text{RKME}})$. However, Figure 2c shows that two models $f_1$ and $f_2$ generate different data distributions $f_1(\mathbf{p}_\tau)$ and $f_2(\mathbf{p}_\tau)$ conditioned on the user prompt distribution $\mathbf{p}_\tau$.

**Remark.** Example 2.1 shows us that overlooking the interplay between images and prompts leads to impossible cases for distinguishing generative models effectively. Existing RKME studies mainly focus on classification tasks, which can implicitly model the tasks through data distribution since the class space is discrete and small. For generative models, we have to explicitly model the model's functionality, i.e., the relation between images and prompts, to achieve satisfied identification results.

**Incorporating relation between images and prompts** Motivated by our analysis, how to incorporate the relationship between images and prompts in model specification and identifying process is the key challenge for our GMI setting. Inspired by existing studies (Li et al., 2015; Ren et al., 2016) about the conditional maximum mean discrepancy, we propose to consider the above relation using a weighted formulation of Equation 2:

$$\mathcal{A}_e^{\text{Weighted}}(S_m^{\text{Weighted}}, R_\tau^{\text{Weighted}}) = \left\| \sum_{i=1}^{N_m} \frac{1}{N_m} w_{m,i} \cdot k(\mathbf{x}_{m,i}, \cdot) - k(\mathbf{x}_\tau, \cdot) \right\|_{\mathcal{H}_k}^2 \tag{3}$$

where $\mathbf{W}_m = \{w_{m,i}\}_{i=1}^{N_m}$ are required to measure the relation between user image $\mathbf{x}_\tau$ and prompt set $\mathbf{P}$. Here, we make the simplifications $R_\tau^{\text{Weighted}} = \mathbf{x}_\tau$ and $S_m^{\text{Weighted}} = \mathbf{X}_m$ in Equation 3. This raises challenges inherent in dimensionality since stable diffusion models produce high-quality images. Moreover, measuring the relation using $\mathbf{W}_m$ is also a challenging problem and encounters cross-modality issues. Below we propose a comprehensive solution based on Equation 3 addressing these challenges.

## 3 PROPOSED METHOD

In this section, we present our solution for the GMI setting, building upon the previous analysis and the weighted formulation introduced in Equation 3. As mentioned earlier, two significant challenges remain to be addressed: 1) The raw images reside in a high-dimensional space, and pixel-level comparisons are highly sensitive. How can we efficiently and effectively measure the similarity between images? 2) The user's image $\mathbf{x}_\tau$ and the platform's prompt set $\mathbf{P}$ belong to different modalities. How do we address cross-modality issues and calculate $\mathbf{W}_m$ to capture the relationship between images and prompts?

To address the aforementioned challenges, we employ a large pre-trained vision model $\mathcal{G}(\cdot)$ to map images from raw input space to a common feature representation space. Subsequently, an image interrogator $\mathcal{I}(\cdot)$ is adopted to convert $\mathbf{x}_\tau$ to corresponding pseudo prompt $\widehat{\mathbf{p}}_\tau$, thereby mitigating the cross-modality issues. Consequently, the similarity in the common feature representation space can be computed with the help of a large pre-trained language model $\mathcal{T}(\cdot)$. We provide a detailed description of our proposed method for the submitting stage and the identification stage, respectively.

### 3.1 SUBMITTING STAGE

In the submitting stage, the model developer submits the generative model $f_m$, and the platform generates the model specification in the background using the specification algorithm $\mathcal{A}_s$ with the submitted models $f_m$ and default prompt set $\mathbf{P}$. The developer can optionally replace $\mathbf{P}$ with a specific prompt set to generate a more precise specification. The algorithm $\mathcal{A}_s$ first samples images from the generative model $f_m$ using the prompt set:

$$\mathbf{X}_m = \{f_m(\mathbf{p})|\mathbf{p} \in \mathbf{P}\} \tag{4}$$

Then, the large pre-trained vision model $\mathcal{G}(\cdot)$ is adopted to encode $\mathbf{X}_m$ as follows. The obtained feature representation $\mathbf{Z}_m$ is efficient and robust to compute similarity between images.

$$\mathbf{Z}_m = \{\mathcal{G}(\mathbf{x})|\mathbf{x} \in \mathbf{X}_m\} \tag{5}$$

Subsequently, $\mathcal{A}_s$ encodes prompt set $\mathbf{P}$ to the common feature representations using $\mathcal{T}(\cdot)$:

$$\mathbf{Q}_m = \{\mathcal{T}(\mathbf{p})|\mathbf{p} \in \mathbf{P}\} \tag{6}$$

Finally, the specification $S_m$ of generative model $f_m$ is defined as follows:

$$S_m = \mathcal{A}_s(f_m; \mathbf{P}_m) = \{\mathbf{Z}_m; \mathbf{Q}_m\} \tag{7}$$

Note that $S_m$ is automatically computed inside the platform, which is very convenient for developers to use and deduce their burden of uploading models. Additionally, the specification does not occupy a large amount of storage space on the platform since the only feature representation is storage.

### 3.2 IDENTIFICATION STAGE

In the identification stage, the users upload one single image $\mathbf{x}_\tau$ to describe their requirements. Then, the platform describes the requirements with $R_\tau$ from $\mathbf{x}_\tau$. Specifically, the requirement algorithm $\mathcal{A}_r$ first generates feature representations of $\mathbf{x}_\tau$ using $\mathcal{G}(\cdot)$, i.e., $\mathbf{z}_\tau = \mathcal{G}(\mathbf{x}_\tau)$. Subsequently, the pseudo-prompt $\widehat{\mathbf{p}}_\tau$ is generated by $\mathcal{I}(\cdot)$, i.e., $\widehat{\mathbf{p}}_\tau = \mathcal{I}(\mathbf{x}_\tau)$, and converted to feature representations using $\mathcal{T}(\cdot)$, i.e., $\widehat{\mathbf{q}}_\tau = \mathcal{T}(\widehat{\mathbf{p}}_\tau)$. The user can optionally replace $\widehat{\mathbf{p}}_\tau$ with a prompt $\mathbf{p}_\tau$ built on his understanding to precisely describe the requirement. Finally, the requirement is:

$$R_\tau = \mathcal{A}_r(\mathbf{x}) = \{\mathbf{z}_\tau; \widehat{\mathbf{q}}_\tau\} \tag{8}$$

Note that $R_\tau$ is automatically computed inside the platform, which is very easy to use for users.

After the platform generates the requirement $R_\tau$, it will calculates the similarity score for each model $f_m$ using evaluation algorithm $\mathcal{A}_e$:

$$\mathcal{A}_e(S_m, R_\tau) = \left\| \sum_{i=1}^{N_m} \frac{1}{N_m} \frac{\widehat{\mathbf{q}}_{m,i}\widehat{\mathbf{q}}_\tau}{\|\widehat{\mathbf{q}}_{m,i}\|\|\widehat{\mathbf{q}}_\tau\|} k(\mathbf{z}_{m,i}, \cdot) - k(\mathbf{z}_\tau, \cdot) \right\|_{\mathcal{H}_k}^2 \tag{9}$$

where the $\mathbf{W}_m$ of Equation 3 is define as the cosine similarity between platform prompts $\widehat{\mathbf{q}}_{m,i} \in \widehat{\mathbf{Q}}_m$ and pseudo-prompt $\widehat{\mathbf{q}}_\tau$. $\mathbf{W}_m$ encodes the structure information of $\mathbf{x}_\tau$ within $\mathbf{P}_m$ during the identification, which successfully captures the relation between images and prompts. The platform returns a list of models sorted in increasing order of similarity score obtained by Equation 9.

## 3.3 DISCUSSION

It is evident that our proposal for the GMI scenario achieves a higher level of accuracy and efficiency when compared to model search techniques employed by existing model hubs.

For accuracy, our proposal elucidates the functionalities of generated models by capturing both the distribution of generated images and prompts. This approach allows for more accurate identification of suitable models for users, as opposed to the traditional model search method that relies on download counts and star ratings for ranking models.

For efficiency, suppose the platform generates one requirement in $T_r$ time and calculates the similarity score for each model in $T_s$ time. The time complexity of our proposal for one identification is $O(T_r + MT_s)$ time. Moreover, with accurate identification results, users can save the efforts of browsing and selecting models, as well as reducing the consumption of network and computing. This is linearly correlated to the number of models on the platform (which can be reduced through tag filtering). Additionally, our approach also has the potential to achieve further acceleration through the use of a vector database (Guo et al., 2023) such as Faiss (Johnson et al., 2019).

## 4 EXPERIMENTS

To verify the effectiveness of our proposed method for GMI problem, we conduct experiments on a novel generative model identification benchmark dataset based on stable diffusion models (Rombach et al., 2022). Our objective is to answer the following three research questions:

- Whether the most suitable generative model can be identified by our proposed method?
- Whether our proposal can achieve satisfactory model recommendations for users?
- To what extent does each component contribute to the proposed method?

## 4.1 EXPERIMENTAL SETTINGS

**Model Platform and Task Construction.**   In practice, we expect model developers to submit their models and corresponding prompts to the model platform. And we expect users to identify models for their real needs. In our experiments, we constructed a model platform and user identification tasks respectively to simulate the above situation. For the construction of the model platform, we manually collect $M = 16$ different stable diffusion models $\{f_1, \ldots, f_M\}$ from one popular model platform, CivitAI, as uploaded generative models on the platform. Note that these collected models belong to the same category to simulate the real process in which users first trigger category filters and then select the models. We construct 55 prompts $\{\mathbf{p}_1, \ldots, \mathbf{p}_{55}\}$ as default prompt set $\mathbf{P}$ of platform. For task construction, we construct 18 evaluation prompts $\{\mathbf{p}_{\tau_1}, \ldots, \mathbf{p}_{\tau_{18}}\}$ for each model on the platform to generate testing images with random seed in $\{0, 1, 2, 3, 4, 5, 6, 7, 8, 9\}$, forming $N_\tau = 18 \times 16 \times 10 = 2880$ different identification tasks $\{(\mathbf{x}_{\tau_i}, t_i)\}_{i=1}^{N_\tau}$, where each testing image $\mathbf{x}_{\tau_i}$ is generated by model $f_{t_i}$ and its best matching model index is $t_i$. Here, we ensure that there is no overlap between $\{\mathbf{p}_1, \ldots, \mathbf{p}_{55}\}$ and $\{\mathbf{p}_{\tau_1}, \ldots, \mathbf{p}_{\tau_{18}}\}$ to ensure the correctness of the evaluation.

**Evaluation Metrics.**   In our experiments, we use accuracy and average rank to evaluate the performance of methods. We define the rank of model $f_m$ for task $\tau$ as $\widehat{r}_{\tau,m} = 1 + \sum_{i=1}^M \mathbb{I}\left[\widehat{s}_{\tau,i} < \widehat{s}_{\tau,m}\right]$. The accuracy is defined as $Acc. = \frac{1}{N^\tau}\mathbb{I}\left[\widehat{r}_{\tau_i, t_i} = 1\right]$, where $Acc. \in [0, 1]$ evaluates the ability of each method to find the best matching model. The average rank is defined as $Rank = \frac{\widehat{r}_{\tau_i, t_i}}{N^\tau}$, where $Rank \in [1, M]$ evaluates the ability of each method to rank the best matching model among other models. We additionally report the $Top\text{-}k$ accuracy, which is calculated as $Top\text{-}k\ Acc. = \frac{1}{N^\tau}\mathbb{I}\left[\widehat{r}_{\tau_i, t_i} \leq k\right]$. This metric measures the average effort spent by users during the identification process and $Top\text{-}k\ Acc. \in [0, 1]$.

**Comparison Methods.**   Initially, we compare it with the traditional model search method called Download. This method is used to simulate how users search generative models according to their downloading volumes (Shen et al., 2023), where users will try models with high downloading volume first. This baseline method can represent a family of methods that employ statistical information

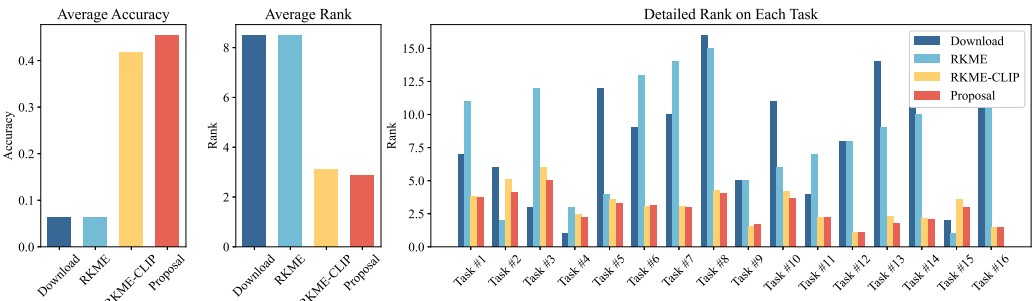

Figure 3: The left two subfigures present the average accuracy and rank on all tasks. The right subfigure presents the detailed rank of each task. For accuracy, the higher the better. For rank, the lower the better. The accuracy shows that our proposal outperforms existing solutions (e.g., Download and RKME) and our simple solution which combines some existing techniques. The average rank and detailed ranks present the user's efforts in identifying satisfied models for each task. Our proposal requires minimal user effort.

without regard to model capabilities. We also consider the basic implementation of the RKME specification (Wu et al., 2023) as a baseline method RKME-Baisc for our GMI problem. The details of generating specifications, and identifying models are presented in section 2.2. Furthermore, we compare our proposed method with a variant of the basic RKME specification, that is, RKME-CLIP, which calculates specifications in the feature representation space encoded by the CLIP model (Radford et al., 2021). The results obtained from RKME-CLIP further support our viewpoint on the critical challenges posed by dimensionality.

**Implementation Details.** We adopt the official code in Wu et al. (2023) to implement the RKME-Basic method and the official code in Radford et al. (2021) to implement the CLIP model. For RKME-Basic and RKME-CLIP methods, we follow the default hyperparameter setting of RKME in previous studies (Guo et al., 2023). We set the size of the reduced set to 1 and choose the RBF kernel (Xu et al., 1994) for RKHS. The hyperparameter $\gamma$ for calculating RBF kernel and similarity score is tuned from $\{0.005, 0.006, 0.007, 0.008, 0.009, 0.01, 0.02, 0.03, 0.04, 0.05\}$ and set to $0.02$ in our experiments. Experiment results below show that our proposal is robust to $\gamma$.

## 4.2 EMPIRICAL RESULTS

**Whether the most suitable generative model can be identified by our proposed method?** The objective of GMI setting is to identify the most suitable generative model for the user's needs. Hence, our initial focus is to determine the effectiveness of baseline methods and our proposed method in identifying the most suitable generative model. We report the accuracy in the left subfigure of Figure 3 to evaluate each method's ability to identify the best matching generative model. Specifically, the Download and RKME-Basic methods cannot work in the GMI problem. The Download method will return models ranked by download volume, which is unable to meet the various needs of users. The identification results of the RKME-Basic method are biased to one model in the platform. The high resolution of images, such as 512x512, presents challenges in calculating the RKME specification and renders the RKME-Basic method ineffective. The performance of RKME-CLIP demonstrates that encoding images is necessary to address the high dimensionality in GMI. However, RKME-CLIP fails to consider the relation between images and prompts, which cannot give the optimal identification results. Our proposal solves the above challenges, giving the best average accuracy compared to baseline methods. These results demonstrate that the specification can help identify the most suitable generative models, which is in line with our starting point.

**Whether our proposal can achieve satisfactory model recommendations for users?** When deploying our proposal in real model platforms, the platform will recommend multiple models for users sorted by similarity score. Therefore, our focus shifts to rank and $Top\text{-}k$ accuracy metrics. We report the average rank and detailed ranks in the right two subfigures of Figure 3. Specifically,

Table 1: Performance of each method evaluated by $Top$-$k$ accuracy. The results show that our proposal achieves 80% top-4 accuracy, indicating that user only requires four models to satisfy their needs in major cases.

| Method | Top-1 Acc. | Top-2 Acc. | Top-3 Acc. | Top-4 Acc. | Top-5 Acc. | Top-6 Acc. | Top-7 Acc. | Top-8 Acc. |
|---|---|---|---|---|---|---|---|---|
| Download | 0.062 | 0.125 | 0.188 | 0.250 | 0.312 | 0.375 | 0.438 | 0.500 |
| RKME-Basic | 0.062 | 0.125 | 0.188 | 0.250 | 0.312 | 0.375 | 0.438 | 0.500 |
| RKME-CLIP | 0.419 | 0.576 | 0.688 | 0.770 | 0.832 | 0.870 | 0.905 | 0.934 |
| Proposal | **0.455** | **0.614** | **0.734** | **0.812** | **0.863** | **0.899** | **0.922** | **0.943** |

Table 2: Ablation study. For accuracy, the higher the better. For rank, the lower the better. The best performance is in bold.

| Methods | Acc. | Top-2 Acc. | Rank |
|---|---|---|---|
| Download | 0.062 | 0.125 | 8.500 |
| RKME-Basic | 0.062 | 0.125 | 8.500 |
| RKME-CLIP | 0.419 | 0.576 | 3.130 |
| RKME-Concat | 0.433 | 0.602 | 2.938 |
| Proposal | **0.455** | **0.614** | **2.852** |

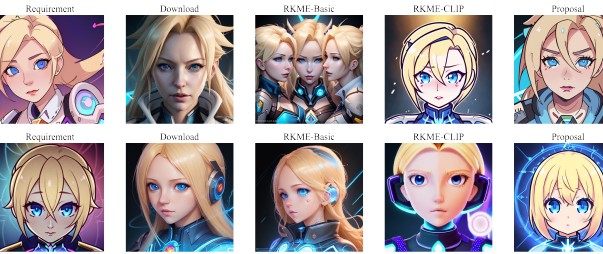

Figure 4: Visualization of generated images.

Download and RKME-Basic methods show poor performance for similar reasons stated above. Our proposal achieves the best average rank, thus demonstrating its effectiveness. Note that the average rank is related to the efforts of the user when identifying the most appropriate models. Therefore, our proposal can save users time and effort compared to baseline methods. We also report the $Top$-$k$ accuracy in Table 1. Our proposal gives the best performance among all different values of K, which demonstrates its effectiveness. The experiment results show that our proposal can achieve 80% accuracy when just recommending four models for users. These results reveal the possibility of building a generative model platform and recommending models to users with specifications.

**To what extent does each component contribute to the proposed method?** In order to comprehensively evaluate the effectiveness of our proposal, we investigate whether each component contributes to the final performance. We additionally compare our proposal with two variants, called RKME-CLIP and RKME-Concat. RKME-CLIP adopts the CLIP model to extract the feature representation for constructing RKME specifications. RKME-Concat adopts both vision and text branches of the CLIP model to extract representations of images and prompts. It combines two modes of representation for constructing RKME specifications. We report accuracy and rank metrics in Figure 2. The performance of RKME-CLIP demonstrates that employing large pre-trained models is an effective approach for addressing dimensionality issues. The performance of RKME-Concat demonstrates the benefits of considering both images and prompts for model identification. Our results achieve the best performance, and demonstrate the effectiveness of our weighted formulation in Equation 3 and our specifically designed algorithm in Equation 9.

## 4.3 VISUALIZATION

We conducted visualization experiments to further show that the new proposal can identify the best-matched model and thereby generate images that meet user requirements better in Figure 4. Specifically, we show the example image $\mathbf{x}_\tau$ submitted by the user in the first column. Then, we generate requirement $R_\tau$ and identify models from the platform using different methods. For each method, we show the image generated with its identified model and pseudo-prompt $\mathcal{I}(\mathbf{x}_\tau)$. The title of the images indicates different model identification methods. The results clearly show that the generative model identified via the new proposal can generate images that best match the users' requirements (most similar to the example image). For example, our method correctly captures the two different comic styles of the image and generates a satisfied image, whereas other methods either have a mismatch in style or have errors in content.

## 5 RELATED WORK

Generative modeling (Jebara, 2012) is a field of machine learning that focuses on learning the underlying distribution and generation of new samples for corresponding distribution. Recently, significant progress has been made in image generation with various methods. Generative Adversarial Networks (GANs) (Arjovsky et al., 2017; Brock et al., 2019; Choi et al., 2020; Goodfellow et al., 2014) apply an adversarial approach to learn the data distribution. It consists of a generator and a discriminator playing a min-max game during the training process. Variational Autoencoders (VAEs) (Kingma & Welling, 2014; Vahdat & Kautz, 2020; van den Oord et al., 2017) is a variant of Auto-Encoder (AE) (Wang et al., 2016), where both consist of the encoder and decoder networks. The encoder in AE learns to map an image into a latent representation. Then, the decoder aims to reconstruct the image from that latent representation. Diffusion Models (DMs) (Nichol & Dhariwal, 2021; Dhariwal & Nichol, 2021; Rombach et al., 2022) leverages the concept of the diffusion process, consisting of forward and reverse diffusion processes. Noise is added to an image during the forward process and the diffusion model learns to denoise and reconstruct the image. With the development of the generative model, various generative model hubs/pools, e.g., HuggingFace, Civitai, have been developed. However, they lack model management and identification mechanisms, resulting in inefficiency for users to find the most suitable model.

Lu et al. (2022) performs context-based search for unconditional generative models and involves a contrastive learning process for all models in the model hubs. However, this learning process significantly hinders the adaptability of the approach, making it unsuitable for a frequently updated model hub. Assessing the transferability of pre-trained models is related to the problem studied in this paper. Negative Conditional Entropy (NCE) (Tran et al., 2019) proposed an information-theoretic quantity (Cover, 1999) to study the transferability and hardness between classification tasks. LEEP (Nguyen et al., 2020) is primarily developed with a focus on supervised pre-trained models transferred to classification tasks. You et al. (2021) designs a general algorithm, which is applicable to vast transfer learning settings with supervised and unsupervised pre-trained models, downstream tasks, and modalities. However, these methods are not suitable for our GMI problem because they impose significant computational overhead in terms of model inference during the identification process. Learnware (Zhou, 2016) presents a general and realistic paradigm by assigning a specification to models to describe their functionalities and utilities, making it convenient for users to identify the most suitable models. Model specification is the key to the learnware paradigm. Recent studies (Tan et al., 2022) are designed on Reduced Kernel Mean Embedding (RKME) (Wu et al., 2023), which achieves model identification by comparing similarities in the RHKS. Tan et al. (2023; 2022) make their efforts to solve heterogeneous feature spaces. However, these studies primarily focus on classification tasks, overlooking the relationship between images and prompts, which is crucial for identifying generative models. Therefore, existing techniques are inadequate for addressing the GMI problem, requiring for novel technologies.

## 6 CONCLUSION

In this paper, for the first time, we propose a novel problem called *Generative Model Identification*. The objective of GMI is to describe the functionalities of generative models precisely and enable the model to be accurately and efficiently identified in the future by users' requirements. To this end, we present a systematic solution including a weighted RKME framework to capture the generated image distributions and the relationship between images and prompts, a large pre-trained vision-language model aimed at addressing dimensionality challenges, and an image interrogator designed to tackle cross-modality issues. Moreover, we built and released a benchmark platform based on stable diffusion models for GMI. Extensive experiment results on the benchmark clearly demonstrate the effectiveness of our proposal. For example, our proposal achieves more than 80% top-4 identification accuracy using just one example image to describe the users' requirements, indicating that users can efficiently identify the best-matched model within four attempts in major cases.

In future work, we will endeavor to develop a novel generative model platform based on the techniques presented in this paper, aiming to provide a more precise description of generative model functionalities and user requirements. This will assist users in efficiently discovering models that align with their specific requirements. We believe this could facilitate the development and widespread usage of generative models.

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

# A ADDITIONAL EXPERIMENTS

## A.1 VISUALIZATION

In this section, we present the additional visualization in Figure 5. Each image is genearated using prompts obtained from image interrogator and models identified by corresponding methods. The

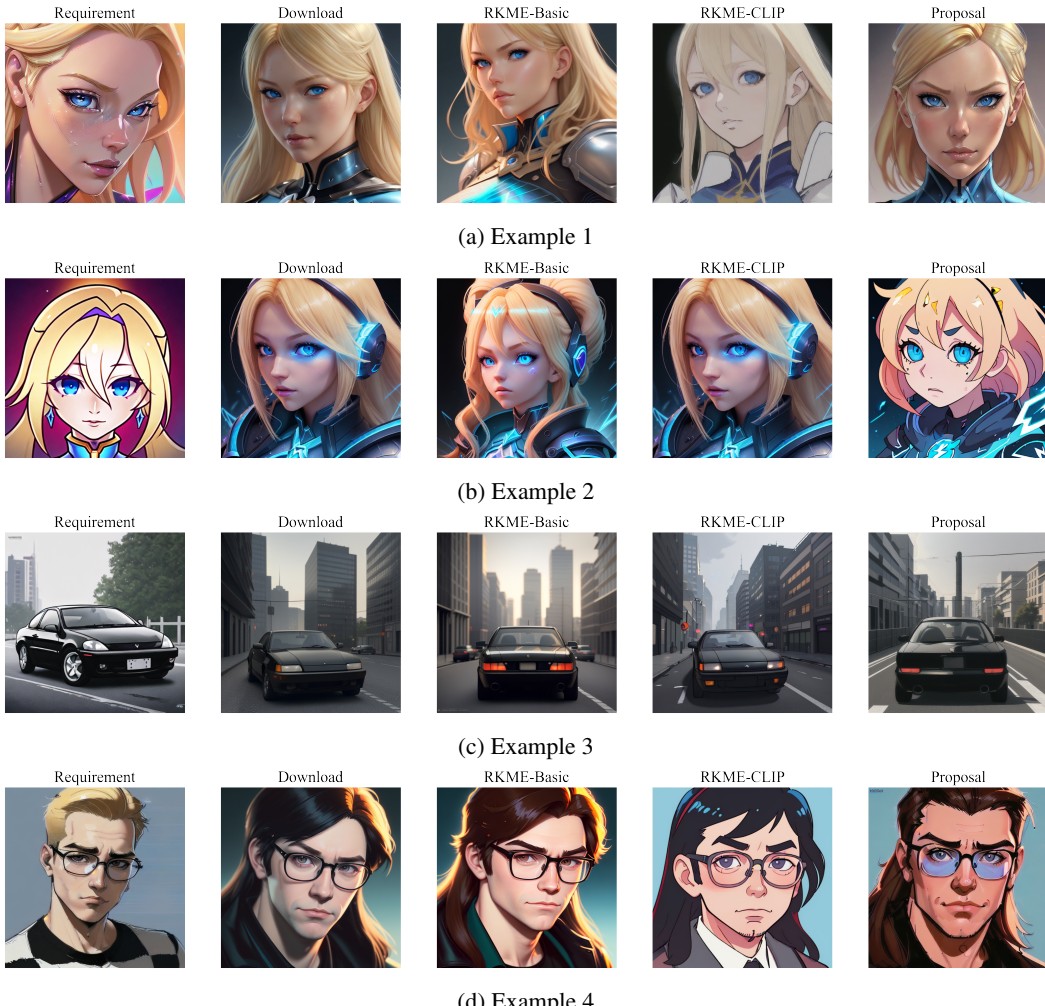

Figure 5: Examples of images generated by the identified model of each method.

results show that our proposal can identify the best-matched model and generate the most similar images, compared to baseline methods. For all examples, the Download method is biased towards the DreamShaper model, while the RKME-basic method is biased towards the RevAnimated model. Our proposed methods successfully identify the most suitable model in examples 1 and 4, while the other RKME-Clip method fails to find the best model. As a result, our proposal yields superior generated results compared to the results of other methods. For example 2, our proposed method identified a model that can generate more cartoonish images. However, the models identified by other methods all generate more realistic images. For example 3, all methods did not find the most suitable model, but the model identified by our method was not inferior to other methods. Detailed results about ground-truth prompts, generated prompts, ground-truth model and identified models are shown in Table 3.

Table 3: The prompts generated using the image interrogator, the corresponding ground-truth prompts, the ground-truth model, and the identified model using different methods for each example.

| | Ground-truth Prompts | Generated Prompts | Ground-truth Model | Download | RKME-basic | RKME-CLIP | Proposal |
|---|---|---|---|---|---|---|---|
| Example 1 | a woman with blonde hair and blue eyes, a detailed painting, by rossdraws, fantasy art, red-purple gradient map, mercy from overwatch, close up of a blonde woman, a brightly colored, lux, sylas | a woman with blonde hair and blue eyes, artgerm. high detail, extremely detailed artgerm, marc brunet, trending artgerm, artgerm on artstation pixiv, featured on artgerm, as seen on artgerm, artgerm comic, artgerm detailed, artgerm. anime illustration | ComicBabes | DreamShaper | RevAnimated | TAnime | ComicBabes |
| Example 2 | a woman with blonde hair and blue eyes, a detailed painting, by rossdraws, fantasy art, red-purple gradient map, mercy from overwatch, close up of a blonde woman, a brightly colored, lux, sylas | a woman with blonde hair and blue eyes, mercy ( overwatch ), overwatch splash art, mercy from overwatch, mercy from overwatch game (2016), heroes of the storm splash art, overwatch fanart, iconic character splash art, official splash art, art of kryssalian, character portrait closeup, cinematic closeup!!, 4 k asymmetrical portrait, zerochan art | QMega | DreamShaper | RevAnimated | DreamShaper | 2DAnimerge |
| Example 3 | a black car parked on the side of the road, a computer rendering, inspired by Stefan Lochner, verdadism, looks like jerma985, on a great neoclassical square, insignia, veveltaria, rowan atkinson, vetements, written in a neat, an ultra realistic 8k octa photo | a black car driving down a street next to tall buildings, ctane 3 d rendered, render in re engine, photorealistic -20, virtual engine 5, 1 9 9 8 render, vue 3d render, rendered in 3 dsmax, denoised photorealistic render, photorealistic highly detailed, photoreal render, cg graphics, v-ray engine | QMega | DreamShaper | RevAnimated | 2DAnimerge | Mistoon |
| Example 4 | a man with glasses and a striped shirt, a picture, inspired by Victor Meirelles, cubo-futurism, 2019 trending photo, clean shaven wide wide wide face, joel fletcher, pixel degradation, uncropped, tired face, k-pop, lolth, tiled, blond boy | a close up of a person wearing glasses, ilya kuvshinov face, ilya kuvshinov with long hair, nerdy man character portrait, portrait of archie andrews, viktor antonov concept art, covid-19 as a human, ilya kuvshinov!, artstyle : ilya kuvshinov, disco elysium character | ComicBabes | DreamShaper | RevAnimated | 2DAnimerge | ComicBabes |

Table 4: Performance when users upload multiple images as their requirements

| | Top-1 Acc. | Top-2 Acc. | Top-3 Acc. | Top-4 Acc. |
|---|---|---|---|---|
| 1 image | 0.455 | 0.614 | 0.734 | 0.812 |
| 2 images | 0.578 | 0.743 | 0.840 | 0.908 |
| 3 images | 0.658 | 0.791 | 0.883 | 0.950 |
| 4 images | 0.709 | 0.854 | 0.929 | 0.969 |
| 5 images | 0.755 | 0.873 | 0.946 | 0.979 |

Here, Image interrogator is implemented with official code[3] for optimizing text prompts to match a given image. It first generates candidate captions using the BLIP model (Li et al., 2022). Then, it adopts a search process to identify the caption list that maximizes the similarity between the captions and images, evaluated by the CLIP model (Radford et al., 2021).

## A.2 MULTIPLE IMAGES FOR QUERYING

RKME paradigm performs more effectively when the provided image set accurately represents the distribution. Thus, our framework, built upon RKME, naturally accommodates situations in which users provide multiple images as queries. We conducted experiments using user queries consisting of 1, 2, 3, 4, or 5 images. In this case, the performance comparison is a bit unfair as lines 2 to 5 involve more images for querying. The results are shown in Table 4. The results demonstrate that the accuracy of identifying the most appropriate model improves as the number of uploaded images increases. Therefore, our framework has the capability to handle multiple images as a query. However, uploading a single image offers a compromise between performance and convenience.

## A.3 DIFFERENT SELECTION OF DEFAULT PROMPT SETS

This section involves conducting experiments to assess the impact of different default prompt sets on our performance. We denote Split 1 as our default prompt set used in our experiments. Split 2 and Split 3 are two non-overlapping subsets, each half the size of Split 1. The results demonstrate the robustness of the identifying performance to varying default prompt sets used in generating specifications.

---

[3]https://github.com/pharmapsychotic/clip-interrogator

Table 5: Performance using different default prompt sets

|         | Top-1 Acc. | Top-2 Acc. | Top-3 Acc. | Top-4 Acc. |
|---------|------------|------------|------------|------------|
| Split 1 | 0.455      | 0.614      | 0.734      | 0.812      |
| Split 2 | 0.442      | 0.609      | 0.729      | 0.810      |
| Split 3 | 0.460      | 0.617      | 0.732      | 0.808      |

Table 6: Performance using an extreme small reduced set size

|           | Top-1 Acc. | Top-2 Acc. | Top-3 Acc. | Top-4 Acc. |
|-----------|------------|------------|------------|------------|
| Size = 1  | 0.434      | 0.592      | 0.707      | 0.789      |
| Proposal  | 0.455      | 0.614      | 0.734      | 0.812      |

## A.4 SCALABILITY PROBLEM

We conduct an extreme experiment with reduced set size = 1 as the size will affect the running time for each query. Compared to proposed methods, the performance is relative stable in Table 6. Therefore, the scalability of the method is guaranteed when the number of generative models increases. As discussed in Section 3.3, scalability problem can be handled using other techniques.

## A.5 DEVELOPER & USER PROVIDED PROMPT

We claim that if the developers and users provide their own prompt, the identify results can be more accurate. The detailed explanations and instructions are given as follows.

Users should provide prompts that accurately describe the images they upload. In our proposed method, we generate prompts by employing an image interrogator that demonstrated effective performance in our experiments. When ground-truth prompts are provided, the identification performance will intuitively not be worse.

Developers should provide prompts to guide the model in generating images it excels at. A model-specific prompt set will enhance identification by amplifying the contrast between generated specifications, resulting in improved accuracy. We conducted synthetic experiments to validate this claim. In this experiment, developers will provide 5 prompts for each model, assuming they excel at them, and generate corresponding specifications. The user will query images generated using similar prompts. The results in Table 7 demonstrate that providing prompts for specification generation leads to improved accuracy. However, in actual situations, it is difficult for the model hub to force developers to provide prompts. Therefore, we conduct experiments under a default prompt set.

## A.6 CONFUSION MATRIX

We present the confusion matrix for the prediction of each method in Figure 6. The Download and RKME algorithm consistently show a bias towards a specific model regardless of the user image $x_\tau$. This indicates that the Download and RKME methods cannot address GMI problem. The results show that our proposal achieves the best identification performance on major tasks.

## A.7 HYPERPARAMETER ROBUSTNESS

We evaluate the robustness of each method to the hyperparameter $\gamma$ in Figure 7. The results demonstrate that our proposed method exhibits robust performance across a wide range of $\gamma$ values. However, as $\gamma$ continues to increase, the performance of both our proposal and the baseline methods begins to degrade. This observation highlights the importance of tuning the hyperparameter $\gamma$ before deploying our method in practical applications. Once $\gamma$ is properly tuned, our method can operate robustly due to its hyperparameter robustness within a broad range.

Table 7: Performance when developers provide specific prompts

|  | Top-1 Acc. | Top-2 Acc. | Top-3 Acc. | Top-4 Acc. |
|---|---|---|---|---|
| Default Prompt Set | 0.455 | 0.614 | 0.734 | 0.812 |
| Developer Provided Prompt | 0.777 | 0.937 | 0.985 | 0.987 |

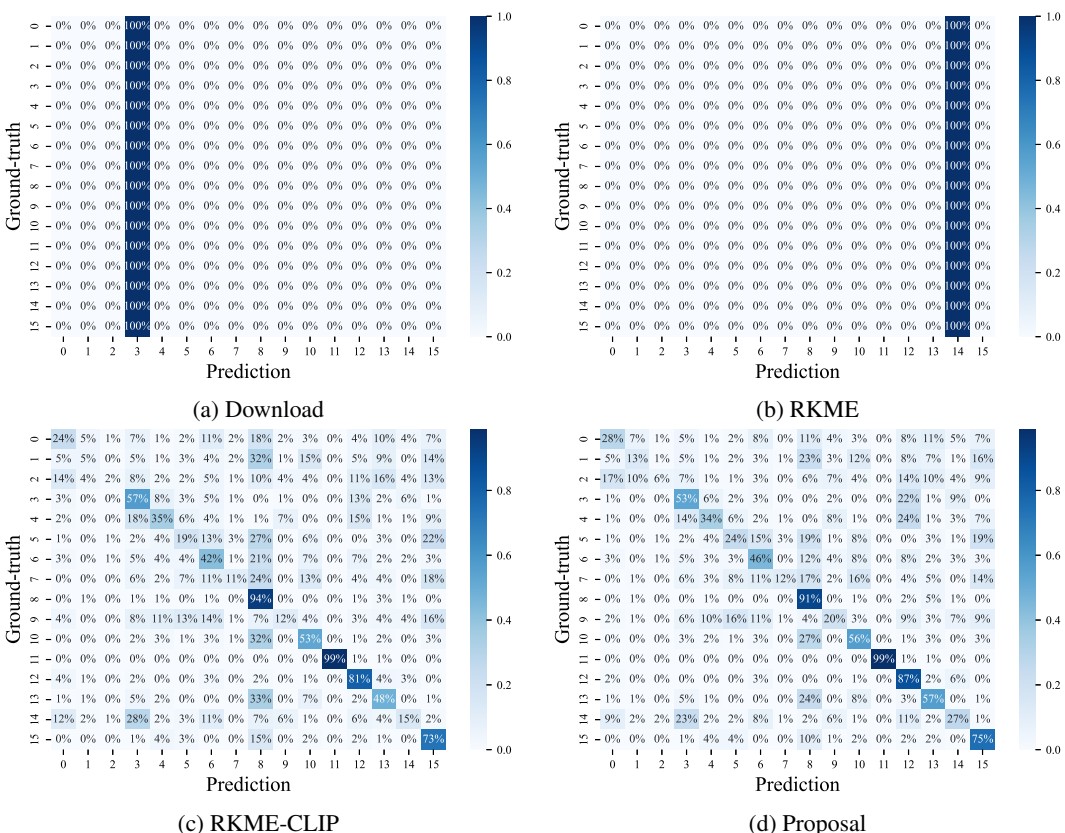

Figure 6: The confusion matrix of each method. The results show that our proposal performs the best. The baseline method, e.g., Download and RKME, cannot work in our GMI problem.

# B  DETAILED EXPLANATION FOR FIGURE 1

In Figure 1, we compare the difference between GMI setting and traditional model search process.

In traditional searches for generative models, the models are initially filtered by the model hubs using tags and then sorted based on download volume. However, this approach presents challenges for users in identifying their desired models, as the model with the highest download volume may not necessarily align with their specific target. Consequently, users select models based on their own judgement, and then download and evaluate each model individually. This consumes a significant amount of network resources and human effort, and there is also the possibility of filtering out the potentially best model due to judgment errors.

In our GMI setting, each model is directly sorted based on its matching score. This means that the most suitable model is recommended at the top, enabling users to effortlessly find their desired models with minimum effort.

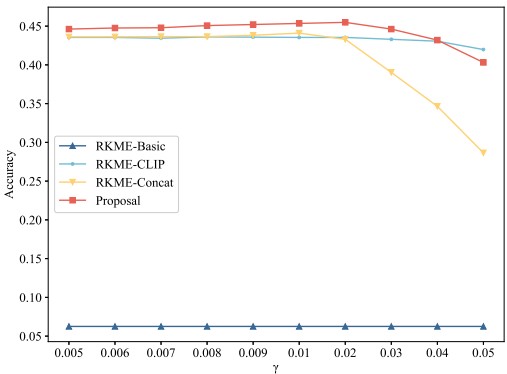

Figure 7: The accuracy with varying values of $\gamma$ was evaluated. The results demonstrate that our proposal is robust to slight changes in the value of $\gamma$.

## C  LIMITATIONS

In this section, we will discuss the limitations of our paper, which encompass two main aspects.

First, the time complexity of our work has the potential for improvement. At present, our time complexity increases linearly with the number of models in the model platform, which will not become a heavy burden when the number of models is large. Previous research (Guo et al., 2023) has demonstrated that acceleration techniques can enhance the identification process, such as the use of a vector database. This could be an area for future exploration.

Second, we currently only consider the cases that identify generative models using one uploaded image to describe users' requirements. The assumption is reasonable since users' ideas often rely on existing image templates when they want to generate images, and it is not difficult to find only one image that has a similar style to fulfill the user's requirements. Despite this, it is also interesting to study how to quickly and accurately identify models via other information such as textual prompts. We will study this problem in future work.

