# OpenReview forum: "You Only Submit One Image to Find the Most Suitable Generative Model"
_ICLR.cc/2024/Conference — Submitted to ICLR 2024_

### Official Review · Reviewer_htir · 2023-10-31

**Soundness:** 3 good
**Presentation:** 3 good
**Contribution:** 3 good
**Rating:** 8
**Confidence:** 4

**Summary:**

The authors propose a novel problem for generative models in this paper, called generative model identification. The goal of this problem is to identify an appropriate text-to-image model for a user’s target image. This problem is driven by the fact that model hubs often filter models based on textual descriptions and download counts, which are not sufficient for capturing user requirements. The authors provide a non-trivial solution to this problem based on vision-language models and RKME methods. The experiment shows that the proposed method achieves strong performance compared to the RKME baseline method.

**Strengths:**

1.	The problem proposed in this paper is very interesting and well-motivated by the fact that current model hubs usually filter models via textual descriptions and download counts. As generative AI becomes more and more mainstream, the proposed problem setting will become increasingly important for the generative community.
2.	The solution of this paper is reasonable since modeling the text-to-image mapping is important for capturing the functionality of generative models and then matching the user requirements. The proposed method makes non-trivial contributions compared to the only one related to the RKME baseline method and gives promising performance.
3.	This paper is well-written and easy to follow. The proposed example clearly demonstrates the importance of modeling text-to-image mapping, which is ignored by baseline methods. The method and experiment parts are clear and well-organized.

**Weaknesses:**

1.	Figure 1 shows that users may use more effort in identifying their target models. However, this figure contains too much information and details. Therefore, a detailed textual explanation should be provided for readers to easily understand this illustration.
2.	The authors constructed the model hub and tasks with 16 stable diffusion models. A brief description should be provided for these models.
3.	Minor problems: (1) The X ticks of the left two sub-figures of Figure 2 are not aligned with the bars; (2) “shows” in the second sentence of section A.2 should be “show”.

**Questions:**

Please refer to the questions in the weakness and answer the following questions:

---

> ### Author Response · Authors · 2023-11-15
>
> **Dear Reviewer htir:**
>
> Thank you for the valuable feedback on our paper. We appreciate the time and effort you have put into reviewing our work and we are grateful for encouraging comments such as *interesting and well-motivated setting*, *reasonable solution,* and *good writing*. We have carefully read your review and addressed your concerns as follows.
>
> **[W1] A detailed textual explanation for Figure 1**
>
> Thank you for your valuable suggestion. We have added the following explanation to our revised paper in Appendix B.
>
> In traditional searches for generative models, the models are initially filtered by the model hubs using tags and then sorted based on download volume. However, this approach presents challenges for users in identifying their desired models, as the model with the highest download volume may not necessarily align with their specific target. Consequently, users select models based on their own judgment, and then download and evaluate each model individually. This consumes a significant amount of network resources and human effort, and there is also the possibility of filtering out the potentially best model due to judgment errors.
>
> In our GMI setting, each model is directly sorted based on its matching score. This means that the most suitable model is recommended at the top, enabling users to effortlessly find their desired models with minimum effort.
>
> **[W2] A brief description of models**
>
> Thank you for your valuable advice. We will add a brief description of each model in our appendix.
>
> **[W3] Minor problems**
>
> Thank you for your valuable advice. We have revised these issues in this paper.

---

> ### Author Response · Authors · 2023-11-18
>
> Dear Reviewer htir,
>
> We have responded to your questions point by point in our reply. It is important for us to know whether our responses have addressed your concerns, and we look forward to receiving your further feedback.
>
> Feel free to reply if you have any further questions or suggestions. Thank you!
>
> Best Regards,
>
> Authors

---

> ### Author Response · Authors · 2023-11-20
>
> Dear Reviewer htir,
>
> We would be happy to answer any further questions you have. Feel free to reply if you have any further questions or suggestions. Thank you!
>
> Best Regards,
>
> Authors

---

### Official Review · Reviewer_7QtA · 2023-10-31

**Soundness:** 2 fair
**Presentation:** 2 fair
**Contribution:** 2 fair
**Rating:** 3
**Confidence:** 4

**Summary:**

This paper introduces a comprehensive solution consisting of three pivotal modules: a weighted Reduced Kernel Mean Embedding
(RKME) framework for capturing the generated image distribution and the relationship between images and prompts, a pre-trained vision-language model aimed at addressing dimensionality challenges, and an image interrogator designed to tackle cross-modality issues.

**Strengths:**

1. The application is interesting. There are many models online and how to identify the needed model efficiently is very important.
2. The proposed method is simple and effective.
3. The paper is overall well-written.

**Weaknesses:**

1. The paper is more like a technical report instead of an academic paper.
2. The technical contributions are limited. It is quite trivial to calculate the distance between the uploaded image/prompt and the existing
images/prompts. The MMD distance is a very naive distance metric, and its robustness is questionable.

**Questions:**

1. Highlight the technical contributions of this paper.
2. Discuss the limitation of the proposed method.

---

> ### Author Response · Authors · 2023-11-15
>
> **Dear Reviewer 7QtA:**
>
> Thank you for the valuable feedback on our paper. We appreciate the time and effort you have put into reviewing our work and we are grateful for encouraging comments such as *interesting application* and *simple*, *effective method,* and *good writing*. We have carefully read your review and addressed your concerns as follows.
>
> **[W1 & W2.1 & Q1] Technical contributions**
>
> Our paper makes technical contributions in three specific aspects:
>
> First, we propose a novel and realistic GMI problem. Generative models have garnered considerable attention, accompanied by the emergence of various generative model platforms. However, users face significant inefficiencies when seeking generative models that satisfy their specific requirements within these platforms, as this necessitates a meticulous browsing and testing process. In this paper, we introduce the Generative Model Identification (GMI) problem, encouraging researchers to develop algorithms describing the capabilities of generative models that enable precise alignment between model functionalities and user requirements, thereby achieving expedited searches in generative model exploration. We posit that such endeavors will substantially propel the advancement of generative models and model platforms.
>
> Second, we take a successful step to address the GMI problem by proposing an effective specification for the generative model. It is noteworthy that the proposal is definitely not a simple combination of existing techniques. For example, we analyze the existing RKME method in Figure 2 and show that the existing method not only can not deal with the mapping between prompts and images which is essential for distinguishing generative models, but also can not address the dimensionality and cross-modality problem which hinders the success of GMI. The results in Table 2 clearly prove that any simple combination of existing methods cannot achieve good performance in GMI, and our proposal provides the best performance evaluated by all metrics. Therefore, it is not fair to consider our proposal as a simple combination of existing techniques.
>
> Third, we construct and release a novel benchmark dataset for the GMI problem, which contains general stable diffusion models. Based on this benchmark dataset, we conduct comprehensive experiments to evaluate the existing baseline methods and proposed methods, and the results demonstrate the effectiveness of our proposed method.  It is expected the dataset could facilitate the development of generative models.
>
> We appreciate your valuable suggestions and will highlight our contributions in our revised paper.
>
> **[W2.1] The MMD distance is a very naive distance metric**
>
> MMD distance is a simple yet effective distance metric actually. For model identification tasks, [1] successfully applies the RKME framework to various image recognition tasks, and [2] applies the RKME framework to both tabular classification and image classification tasks. Additionally, MMD distance is widely used in Domain Adaptation [3] and Domain Generalization [4]. Therefore, the MMD distance is not such a naive and questionable distance metric, and our proposed methods are non-trivial based on it.
>
> Moreover, we should clarify that our proposal goes beyond a simple application of RKME and MMD distances. We analyze the existing RKME method in Figure 2 and show that the existing method not only can not deal with the mapping between prompts and images which is essential for distinguishing generative models, but also can not address the dimensionality and cross-modality problem which is a hinder of GMI. The results in Table 2 clearly prove that any simple combination of existing methods cannot achieve good performance in GMI, and our proposal provides the best performance evaluated by all metrics. Therefore,  our proposal is clearly not  a simple combination of existing techniques.
>
> [1] Lan-Zhe Guo, Zhi Zhou, Yu-Feng Li, Zhi-Hua Zhou: Identifying Useful Learnwares for Heterogeneous Label Spaces. ICML 2023: 12122-12131
>
> [2] Yu-Jie Zhang, Yu-Hu Yan, Peng Zhao, Zhi-Hua Zhou: Towards Enabling Learnware to Handle Unseen Jobs. AAAI 2021: 10964-10972
>
> [3] Wei Wang, Haojie Li, Zhengming Ding, Feiping Nie, Junyang Chen, Xiao Dong, Zhihui Wang: Rethinking Maximum Mean Discrepancy for Visual Domain Adaptation. IEEE Trans. Neural Networks Learn. Syst. 34(1): 264-277 (2023)
>
> [4] Peifeng Tong, Wu Su, He Li, Jialin Ding, Zhan Haoxiang, Song Xi Chen: Distribution Free Domain Generalization. ICML 2023: 34369-34378

---

> ### Author Response · Authors · 2023-11-15
>
> **[Q2] Discuss the limitations of the proposed method.**
>
> We have discussed the limitation of efficiency in "3.3 Discussion" in our paper. Thank you for your valuable suggestion. We have additionally added a "Limitations" section in Appendix. C to discuss the limitations of the proposed method.
>
> First, the time complexity of our work has the potential for improvement. At present, our time complexity increases linearly with the number of models in the model platform, which will not become a heavy burden when the number of models is large. Previous research[1] has demonstrated that acceleration techniques can enhance the identification process, such as the use of a vector database. This could be an area for future exploration.
>
> Second, we currently only consider the cases that identify generative models using one uploaded image to describe users' requirements. The assumption is reasonable since users' ideas often rely on existing image templates when they want to generate images, and it is not difficult to find only one image that has a similar style to fulfill the user's requirements. Despite this, it is also interesting to study how to quickly and accurately identify models via other information such as textual prompts. We will study this problem in future work.
>
> [1] Lan-Zhe Guo, Zhi Zhou, Yu-Feng Li, Zhi-Hua Zhou: Identifying Useful Learnwares for Heterogeneous Label Spaces. ICML 2023: 12122-12131

---

> ### Author Response · Authors · 2023-11-18
>
> Dear Reviewer 7QtA,
>
> We have responded to your questions point by point in our reply. It is important for us to know whether our responses have addressed your concerns, and we look forward to receiving your further feedback.
>
> Feel free to reply if you have any further questions or suggestions. Thank you!
>
> Best Regards,
>
> Authors

---

> ### Author Response · Authors · 2023-11-20
>
> Dear Reviewer 7QtA,
>
> We would be happy to answer any further questions you have. If you do not have any further questions, we hope that you might consider raising your score.
>
> Best Regards,
>
> Authors

---

> ### Author Response · Authors · 2023-11-23
>
> Dear Reviewer 7QtA,
>
> Thank you for your time and effort in reviewing our paper.
>
> As the discussion period nears its end, we'd like to highlight that only a few hours remain before the deadline. We assure you that we've addressed your comments and included additional revisions. If you have any remaining concerns, please reach out in the next few hours. Your insights are crucial to us. If you do not have any further questions, we hope that you might consider raising your score.
>
> Best Regards,
>
> Authors

---

### Official Review · Reviewer_pBNB · 2023-10-31

**Soundness:** 3 good
**Presentation:** 2 fair
**Contribution:** 2 fair
**Rating:** 3
**Confidence:** 3

**Summary:**

The paper introduces the GMI (Generative Model Identification) task, aiming to match user-provided images to the best generative model. The process has two stages: first, it precomputes the specification for each generative model (map to feature and compute ; second, it compares the query image's specification with these precomputed specifications using weighted RKME to gauge similarity.

**Strengths:**

- The problem setting addresses a practical need to identify the most suitable generative model amidst a myriad of options we have nowadays.
- The application of RKME as a similarity metric is interesting. There's potential relevance to the "Informative Features for Model Comparison" work, even though the latter primarily focuses on comparing just two models based on the goodness of fit with image queries.
- Separating precomputation and actual comparison is not new, but is a useful concept to speed up the process.

**Weaknesses:**

- The paper assumes that users will always provide an example image, which might not be universally applicable or intuitive.
- The work heavily relies on existing methods: RKME, a pre-trained vision-language model (possibly CLIP?), and the image interrogator from a GitHub repository. There's a lack of novelty in the proposed method.
- Perhaps it's just me, but I found the writing in the technical sections really confusing.
-  While the image interrogator is referenced from an existing work, it would still benefit from a brief description within this paper, given that not all readers may be familiar with it.
- The lack of mention or comparison with the 'Content-Based Search for Deep Generative Models' paper. That work seems to have broader capabilities, being able to process not just image but also text, sketches, or a combination of them. In comparison, the proposed method feels like a less capable variant. Also for the experiment, that paper uses 100+ models whereas this paper uses only 16 models.

**Questions:**

- How was the default prompt 'P' determined?
- The paper suggests that developer/user providing the exact prompt would yield better results. Can this claim be supported with any qualitative or quantitative evaluation? Also, are there any guidelines or recommendations for the type of prompt to be used?

---

> ### Author Response · Authors · 2023-11-15
>
> **Dear Reviewer pBNB:**
>
> Thank you for the valuable feedback on our paper. We appreciate the time and effort you have put into reviewing our work and we are grateful for encouraging comments such as *practical setting* and *the interesting idea*. We have carefully read your review and addressed your concerns as follows.
>
> **[W1] Assumption on providing an example image**
>
> First of all, we need to point out that in the case that users cannot provide additional information, they can only browse and search one by one through the tags and illustration examples of the models in the model hub, which results in users not being able to quickly and accurately search for the models that satisfy their requirements. To realize a fast and accurate search, it is necessary to assume that users should provide additional information to describe their requirements. In this paper, we consider the cases that identify generative models with only one uploaded image to describe users' requirements. The assumption is reasonable since users' ideas often rely on existing image templates when they want to generate images, and it is not difficult to find only one image that has a similar style to the user's requirements. Moreover, it is also interesting to study how to quickly and accurately identify models via other information such as textual prompts, we will study this problem in future work.
>
> **[W2] Lack of novelty in the proposed method**
>
> We thank you for your review. It looks like you have some misconceptions about the contribution of our paper.
>
> First, we propose a novel and realistic GMI problem. Generative models have garnered considerable attention, accompanied by the emergence of various generative model platforms. However, users face significant inefficiencies when seeking generative models that satisfy their specific requirements within these platforms, as this necessitates a meticulous browsing and testing process. In this paper, we introduce the Generative Model Identification (GMI) problem, encouraging researchers to develop algorithms describing the capabilities of generative models that enable precise alignment between model functionalities and user requirements, thereby achieving expedited searches in generative model exploration. We posit that such endeavors will substantially propel the advancement of generative models and model platforms.
>
> Second, we take a successful step to address the GMI problem by proposing an effective specification for the generative model. It is noteworthy that the proposal is definitely not a simple combination of existing techniques. For example, we analyze the existing RKME method in Figure 2 and show that the existing method not only can not deal with the mapping between prompts and images which is essential for distinguishing generative models, but also can not address the dimensionality and cross-modality problem which hinders the success of GMI. The results in Table 2 clearly prove that any simple combination of existing methods cannot achieve good performance in GMI, and our proposal provides the best performance evaluated by all metrics. Therefore, it is not fair to consider our proposal as a simple combination of existing techniques.
>
> Third, we construct and release a novel benchmark dataset for the GMI problem, which contains general stable diffusion models. Based on this benchmark dataset, we conduct comprehensive experiments to evaluate the existing baseline methods and proposed methods, and the results demonstrate the effectiveness of our proposed method.  It is expected the dataset could facilitate the development of generative models.
>
> We will highlight our contributions in our revised paper.
>
> **[W3] Writing problem**
>
> Thank you for your advice. We will improve the writing for technical sections.
>
> **[W4] A brief description of image interrogator**
>
> Thank you for your advice. We have incorporated the following description into our revised paper in Appendix A.1.
>
> The Image Interrogator tool combines CLIP and BLIP to optimize text prompts for matching a given image. Firstly, it generates candidate captions using the BLIP model. Next, it employs a search process to identify the caption list that maximizes the similarity between the captions and images, as evaluated by the CLIP model.

---

> ### Author Response · Authors · 2023-11-15
>
> **[W5] Comparison with the 'Content-Based Search for Deep Generative Models'**
>
> The setting in 'Content-Based Search for Deep Generative Models' (referred to as CBS) differs from ours; therefore, it cannot be directly compared with our paper. We now highlight two key differences:
>
> First, CBS addresses the search problem for unconditional generative models. These models generate images for a predefined concept and style, which can not set a specific prompt. Our paper, instead, concentrates on conditional generative models that sample images under the guidance of a prompt, a methodology that is more frequently employed by models in the model hubs. This difference yields our main challenge to model the mapping between images and prompts, which we discussed in 2.2 Problem Analysis and 3. Proposed Method. This difference also determines that the model zoo of CBS is large because their models only generate one concept in one style each. The models in our paper can generate diverse concepts under the guidance of prompts. Therefore, our model zoo is already large enough to demonstrate the effectiveness of our proposed method.
>
> Secondly, CBS assumes that all models in the platform are known in advance, and employs a contrastive learning process to acquire the matching functions. This limitation significantly impedes the adaptability of the approach, as models within the platform are frequently being added and updated. It is impossible to re-train the matching function each time models are added or updated.
>
> Hence, our paper addresses a more complex and realistic setting where the technique employed in CBS cannot be directly applied. We highly appreciate your valuable suggestion and have incorporated a discussion on 'Content-Based Search for Deep Generative Models' in our related work.
>
> **[Q1] Default prompt 'P'**
>
> This paper focuses on improving the identification of the most suitable model when prompts are already provided. Consequently, we utilize a default set of prompts for all generative models during the construction of specifications. This default set of prompts can be found in the "PromptPool.json" file provided in our supplementary material.
>
> **[Q2] Developer / user provided  prompt**
>
> Thank you for your constructive suggestions.
>
> Users should provide prompts that accurately describe the images they upload. In our proposed method, we generate prompts by employing an image interrogator that demonstrated effective performance in our experiments. When ground-truth prompts are provided, the identification performance will not be worse, and this is supported by our response to Reviewer BSis [Q3].
>
> Developers should provide prompts to guide the model in generating images it excels at. A model-specific prompt set will enhance identification by amplifying the contrast between generated specifications, resulting in improved accuracy. We conducted synthetic experiments to validate this claim. In this experiment, developers will provide 5 prompts for each model, assuming they excel at them, and generate corresponding specifications. The user will query images generated using similar prompts. The results demonstrate that providing prompts for specification generation leads to improved accuracy. However, in actual situations, it is difficult for the model hub to force developers to provide prompts. Therefore, this paper conducts experiments under a default prompt set.
>
> |                             | Top-1 Acc | Top-2 Acc | Top-3 Acc | Top-4 Acc |
> | --------------------------- | --------- | --------- | --------- | --------- |
> | Default Prompt Set          | 0.455     | 0.614     | 0.734     | 0.812     |
> | Developer  Provided  Prompt | 0.777     | 0.937     | 0.985     | 0.987     |
>
> We greatly appreciate the valuable suggestion provided by the reviewer, and we have incorporated this discussion and experiment into our revised paper in Appendix A.5.

---

> ### Author Response · Authors · 2023-11-18
>
> Dear Reviewer pBNB,
>
> We have responded to your questions point by point in our reply. It is important for us to know whether our responses have addressed your concerns, and we look forward to receiving your further feedback.
>
> Feel free to reply if you have any further questions or suggestions. Thank you!
>
> Best Regards,
>
> Authors

---

> ### Author Response · Authors · 2023-11-20
>
> Dear Reviewer pBNB,
>
> We would be happy to answer any further questions you have. If you do not have any further questions, we hope that you might consider raising your score.
>
> Best Regards,
>
> Authors

---

> ### Author Response · Authors · 2023-11-23
>
> Dear Reviewer pBNB,
>
> Thank you for your time and effort in reviewing our paper.
>
> As the discussion period nears its end, we'd like to highlight that only a few hours remain before the deadline. We assure you that we've addressed your comments and included additional revisions. If you have any remaining concerns, please reach out in the next few hours. Your insights are crucial to us. If you do not have any further questions, we hope that you might consider raising your score.
>
> Best Regards,
>
> Authors

---

### Official Review · Reviewer_BSis · 2023-11-01

**Soundness:** 3 good
**Presentation:** 4 excellent
**Contribution:** 2 fair
**Rating:** 6
**Confidence:** 2

**Summary:**

This paper presents generative model identification that can rank a given set of diffusion based generative models that could potentially produce a user provided image. The idea is to represent each generative model using a small number of images and their generative prompts, which are then used in a reduced kernel mean embedding (RKME) scoring function to rank the proximity of the embeddings of the given user provided image in a suitable RKHS. The paper also explores incorporating the prompt in the RKME scoring function using neura embeddings of the prompts and the images. Experiments are provided on a small scale dataset and show promise.

**Strengths:**

1. The problem setting appears novel and useful. Especially, when there are hundreds of models in a model zoo and one needs to find what model could be used to produce a given image.
2. The method is straightforward, the exposition is very clear, and easy to read.
3. Experiments show promising results against existing retrieval baselines.

**Weaknesses:**

1. As I understand, RKME scoring would work better if you have image distributions to match; that is, if you have more than one user provided query image. In that sense, it is not clear to me how the paper could claim that a single image is sufficient to identify the model? How do you substantiate this claim? One thought is: to derive a more general scoring function (in Eq. 2, for example) with multiple user query images and the authors did an ablation that shows that using a single image or when using multiple images, the retrieval performance is approximately similar. Further to the above comment, given there is only a single image query, what is the performance if you used a different statistic for the scoring? For example, using \ell_1 norm or just the \argmin on the kernel features?

2. As the selection of the reduced image sets is important for efficiency of the method, the paper must provide details of how this is done for generative models. How do you ensure the size is large enough and also discriminative for each model so that overlap between reduced sets across models is minimal while the model representation is sufficient for identification?

3. One of the key innovations in this paper is the inclusion of prompt embeddings in the model specification (Eq. 9). As is clear from Eq. 9, the weighting scheme soft-selects images in the reduced set using the query prompt and compares the kernel embeddings of the respective images with the query image. A natural question that would arise here is: how do you ensure the reduced set prompts are selected so as to cover user provided images or prompts? This part seems less detailed and not evaluated in the experiments. Specifically, how do you produce the (prompt, generated image) pairs for the reduced set? How do you ensure the prompts are different for diverse generated images? How do you account for the domain gaps (if any) between user specified prompts against the submission-time specified prompts? What is the sensitivity of the scoring function against differences in the prompts? It would be useful to  see qualitative results showing images and (varied) prompts.

**Questions:**

Other questions.
1. How does the performance of the method vary when using varied reduced set sizes? It is also important to analyze the scalability of the method when the number of generative models goes higher.
2. What are the prompts associated with the qualitative results in Figure 4? Can you also include images with user specified prompts?
3. Why is the performance of RKME-CLIP so close to that of the proposed method? What is the size of the reduced set for the various diffusion models?
4. What would be the performance of the model if: i) the query image was selected from one of the generative models, but captioned using I(.)? ii) the query image was selected from one of the models and with the respective prompts? iii) you use the prompt weight w to be an indicator function when using i). These experiments will show the upper-bound on performance of the method (when using kernel mean matching).
5. What is \gamma in Figure 7?
6. In Figure 6, Examples 2,3, it is not clear to me why the proposed result is better than other ones. An explanation would help.

**Details Of Ethics Concerns:**

No ethical concerns.

---

> ### Author Response · Authors · 2023-11-15
>
> **Dear Reviewer BSis:**
>
> Thank you for the valuable feedback on our paper. We appreciate the time and effort you have put into reviewing our work and we are grateful for encouraging comments such as *the novel and useful setting* and *promising results*. We have carefully read your review and addressed your concerns as follows.
>
> **[W1.1] Multiple images for querying**
>
> RKME performs more effectively when the provided image set accurately represents the distribution. Thus, our framework, built upon RKME, naturally accommodates situations in which users provide multiple images as queries. We conducted experiments using user queries consisting of 1, 2, 3, 4, or 5 images. The results are shown in the following table:
>
> |                | Top-1 Acc | Top-2 Acc | Top-3 Acc | Top-4 Acc |
> | -------------- | --------- | --------- | --------- | --------- |
> | 1 image (Ours) | 0.455     | 0.614     | 0.734     | 0.812     |
> | 2 images       | 0.578     | 0.743     | 0.840     | 0.908     |
> | 3 images       | 0.658     | 0.791     | 0.883     | 0.95      |
> | 4 images       | 0.709     | 0.854     | 0.929     | 0.969     |
> | 5 images       | 0.755     | 0.873     | 0.946     | 0.979     |
>
> The results demonstrate that the accuracy of identifying the most appropriate model improves as the number of uploaded images increases.
>
> Our claim regarding a single image is grounded in two factors: (1) In practical applications, it is often more convenient to upload a single image rather than multiple images to describe the user's requirements; (2) Using one image can achieve a satisfactory Top-4 Accuracy for identifying the most appropriate model.
>
> In conclusion, our framework has the capability to handle multiple images as a query. However, uploading a single image offers a compromise between performance and convenience. We appreciate the reviewer's valuable suggestion and have added this experiment and discussion to our revised paper in Appendix A.2.
>
> **[W1.2] Different statistics for scoring**
>
> We thank you for your constructive advice. We conduct experiments using some statistical scoring functions, e.g., L1, and Min functions. The results are shown in the following table:
>
> |      | Top-1 Acc | Top-2 Acc | Top-3 Acc | Top-4 Acc |
> | ---- | --------- | --------- | --------- | --------- |
> | Min  | 0.064     | 0.116     | 0.163     | 0.233     |
> | L1   | 0.346     | 0.540     | 0.657     | 0.735     |
> | Ours | 0.455     | 0.614     | 0.734     | 0.812     |
>
> The results show that our proposed method can outperform these statistical scoring functions.
>
> **[W2 & W3.1] The selection of the reduced image sets**
>
> We thank you for your professional advice. Selecting suitable prompts is important for building specifications for generative models.
>
> This paper focuses on improving the identification of the most suitable model when prompts are already provided. Consequently, we utilize a default set of prompts for all generative models during the construction of specifications. This default set of prompts can be found in the "PromptPool.json" file provided in our supplementary material. And, our proposal can achieve satisfactory performance when using the default prompt set for all models. We have also conducted experiments on the following questions, e.g., [W3.2], [Q1], [Q4], to answer your questions.
>
> We appreciate the reviewer pointing out these key points for GMI setting. The selection of prompts for generating specifications could be a future direction for the GMI setting.
>
> **[W3.2] Qualitative results for selection of the reduced image sets**
>
> We conducted experiments with three different prompt sets. Split 1 is our default prompt set used in our experiments. Split 2 and Split 3 are two non-overlapping subsets, each half the size of Split 1. The results are shown in the following table:
>
> |                | Top-1 Acc | Top-2 Acc | Top-3 Acc | Top-4 Acc |
> | -------------- | --------- | --------- | --------- | --------- |
> | Split 1 (Ours) | 0.455     | 0.614     | 0.734     | 0.812     |
> | Split 2        | 0.442     | 0.609     | 0.729     | 0.810     |
> | Split 3        | 0.460     | 0.617     | 0.732     | 0.808     |
>
> The results demonstrate the robustness of the identifying performance to varying default prompt sets used in generating specifications. We appreciate the reviewer's valuable suggestion and have added this experiment to our revised paper in Appendix A.3.

---

> ### Author Response · Authors · 2023-11-15
>
> **[Q1] Performance with varied reduced set sizes**
>
> We conduct an extreme experiment with a reduced set size = 1. Compared to our proposed method, which set the reduced set size to 550, the performance is relatively stable and still outperforms comparison methods. Therefore, the scalability of the method is guaranteed when the number of generative models increases.
>
> |                    | Top-1 Acc | Top-2 Acc | Top-3 Acc | Top-4 Acc |
> | ------------------ | --------- | --------- | --------- | --------- |
> | Size = 1 (Ours)    | 0.434     | 0.592     | 0.707     | 0.789     |
> | Size =  550 (Ours) | 0.455     | 0.614     | 0.734     | 0.812     |
>
> As discussed in “3.3.Discussion”, the scalability problem can be handled using other techniques. We appreciate the reviewer's valuable suggestion and have added this experiment to our revised paper in Appendix A.4.
>
> **[Q2] Prompts associated with the qualitative results in Figure 4**
>
> The ground-truth prompt for requirement is `a woman with blonde hair and blue eyes, a detailed painting, by rossdraws, fantasy art, red-purple gradient map, mercy from overwatch, close up of a blonde woman, a brightly colored, lux, sylas`. The prompt generated by image interrogator is `a girl with blonde hair and blue eyes, game icon stylized, glowwave girl portrait, epic legends game icon, app icon, 3 d icon for mobile game, icon for an ai app, darkness background, ios app icon, lux from league of legends, darkness's background, character icon, luxanna crownguard, lightning mage spell icon, darkness aura`. The images generated by each method utilize prompts obtained from the image interrogator. We have added ground-truth prompts and generated prompts for each example in Appendix A.1.
>
> **[Q3] Performance of RKME-CLIP**
>
> RKME-CLIP is an ablation variant of the proposed method that addresses the dimensionality challenge by incorporating the CLIP model. Furthermore, our proposal enhances performance by taking into account the mapping between prompts and images.
>
> **[Q4] Three variants of proposed method**
>
> We conducted the experiments to evaluate two variants and our proposed method in the following table. Method (i) is our proposed method, which adopts the prompt generated from the image interrogator. Method (ii) adopts the ground-truth prompts of query images to calculate the weights. Method (iii) adopts an indicator function instead of soft weights.
>
> |            | Top-1 Acc | Top-2 Acc | Top-3 Acc | Top-4 Acc |
> | ---------- | --------- | --------- | --------- | --------- |
> | (i) (Ours) | 0.455     | 0.614     | 0.734     | 0.812     |
> | (ii)       | 0.455     | 0.615     | 0.733     | 0.813     |
> | (iii)      | 0.252     | 0.371     | 0.461     | 0.534     |
>
> The results show that methods (i) and (ii) perform similarly, indicating that prompts generated from an image interrogator can serve as a proxy for the ground truth when identifying the model. Method (iii) performs less effectively compared to the other two methods, suggesting the importance of soft weights in our proposal.
>
> **[Q5] Gamma in Figure 7**
>
> Gamma serves as a hyper-parameter in the calculation of RKME. The results demonstrate the robustness of our proposal to variations in this hyper-parameter.
>
> **[Q6] Explanations for Examples 2, 3 in Figure 5**
>
> Thank you for your valuable advice. We have added explanations for examples to our revised paper in Appendix A.1.
>
> For example 2, our proposed method identified a model that can generate more cartoonish images. However, the models identified by other methods all generate more realistic images. For example 3, all methods did not find the most suitable model, but the model identified by our method was not inferior to other methods.

---

> ### Author Response · Authors · 2023-11-18
>
> Dear Reviewer BSis,
>
> We have responded to your questions point by point in our reply. It is important for us to know whether our responses have addressed your concerns, and we look forward to receiving your further feedback.
>
> Feel free to reply if you have any further questions or suggestions. Thank you!
>
> Best Regards,
>
> Authors

---

> ### Author Response · Authors · 2023-11-20
>
> Dear Reviewer BSis,
>
> We would be happy to answer any further questions you have. If you do not have any further questions, we hope that you might consider raising your score.
>
> Best Regards,
>
> Authors

---

### Meta-Review · Area_Chair_9AXz · 2023-12-07

**Metareview:**

(a) Scientific claims and findings

This paper delves into the generative model identification problem, focused on discovering a generative model based on user-specified requirements indicated by a visual example. The authors utilize a weighted reduced kernel mean embedding (RKME) as the distance function between models and visual examples during model search. These weights specifically capture the relationship between text prompts and images within text-to-image generative models. Empirical results on a simulation dataset generated by the authors demonstrate the efficacy of the proposed method over the naive, unweighted RKME.

(b) Strength

The problem is both novel and interesting, while the method remains simple yet effective.

(c) Weakness

i. The problem definition is problematic. The assumption that a user can consistently provide a single visual example to specify all requirements is not valid.

ii. An essential aspect of the approach has been overlooked—the construction of the reduced sets, namely, text prompts and generated images for RKME estimation, remains unaddressed. Additionally, the paper does not account for managing the potential domain gap between these reduced sets and user-provided inputs.

iii. The experiment design lacks detailed information, and the experiments were conducted on a notably small-scale dataset compared with related works.

iv. The writing can be improved, particularly the problem and approach sections.

v. The novelty and technical contribution appear limited in this work. It heavily relies on an existing method with minor modifications, and there seems to be inadequate comparison or discussion concerning some highly related works.

**Justification For Why Not Higher Score:**

The rebuttal doesn't resolve the concerns raised by the reviewers.

1. The assumption regarding user requirements being solely specified by an image lacks proper justification.
2. The argument asserting the sufficiency of a default prompt set to represent all models lacks persuasion and justification.
3. The argument that related prior efforts cannot be applied lacks persuasiveness as they remain applicable in the proposed evaluation.
4. There is no provided plan for improving writing and furnishing experiment details.

**Justification For Why Not Lower Score:**

N/A

---

### Decision · Program_Chairs · 2024-01-16

Reject